# Spatio-temporal heterogeneity of China's import and export trade, factors influencing it, and its implications for developing countries' trade

Haisong Wang[ID][1◉], Yuhuan Wu[ID][1◉]*, Ning Zhu[2◉]

**1** Department of Economics and Trade, Hebei University of Water Resources and Electric Engineering, Cangzhou, Hebei, China, **2** Institute of Agricultural Economics and Development, Chinese Academy of Agricultural Sciences, Haidian, Beijing, China

◉ These authors contributed equally to this work.
* wuyuhuan@hbwe.edu.cn

**Data Availability Statement:** All relevant data are within the manuscript and its Supporting information files.

## Abstract

This study constructed a multidimensional indicator system to evaluate spatio-temporal heterogeneity of China's import and export trade of 31 provinces from 2000 to 2022. This study describes the distribution of China's import and export trade by using location Gini coefficient and exploratory spatial analysis. Additionally, Multiple linear regression was used to ascertain the extent of contribution by various factors on the spatio-temporal heterogeneity of import and export trade. The simulation results show that inter-provincial import and export trade displayed distinct spatio-temporal differentiation characteristics with a prominent east-to-west disparity from 2000 to 2022. The trade links between various regions of the country have gradually strengthened, with a corresponding high correlation to the level of economic development. GDP, financial expenditure, freight transportation volume, technology market turnover, foreign investment, and disposable income of all residents, significantly influence the per capita export and import volume. In general, it is suggested that China and developing countries should take effective measures to promote balanced trade development, strengthen regional cooperation and coordination, and promote green trade and sustainable development.

## Introduction

In the context of global economic integration, China has experienced rapid development over the last few decades, emerging as one of the world's leading trading nations [1]. Instead of advancing in a linear fashion, China's import and export trade is, nonetheless, influenced by various temporal and spatial factors [2]. The growth rate of China's import and export trade displays significant variations across different periods. While China's import and export trade experienced a significant impact which resulted in a sharp decline in both export and import during the global financial crisis [3], it rapidly recovered and continued to grow after that.

**Funding:** Supported by Ministry of Education, Industry-University Cooperation Collaborative Education Project(230825052507181). -Funded by Science Research Project of Hebei Education Department (BJS2024097). -Supported by Hebei Province Social Science Development Research Project (20230303051).

**Competing interests:** The authors have declared that no competing interests exist.

This temporal heterogeneity indicates that various factors have driven China's import and export trade at different times. Furthermore, China's import and export trade exhibit inter-regional differences [4, 5]. For instance, China's import and export trade is predominantly dominated by the eastern coastal region, while the western inland region comparatively plays less roles in import and export activities. This spatial distribution disparity is primarily caused by factors such as location, transportation and communication infrastructure, and industrial structure [6].

Further research on the spatio-temporal heterogeneity of China's import and export trade needs to focus on a number of factors. First, China's monetary and exchange rate policies have an important impact on import and export trade [7, 8]. Changes in the exchange rate of the RMB may result in fluctuations in import and export prices, which could impact China's international trade competitiveness [9]. Additionally, foreign direct investment (FDI) significantly influences China's import and export trade [10, 11]. FDI brings capital, technology, and management experience, which in turn promotes China's import and export trade growth. Furthermore, trade friction is another significant factor affecting China's import and export trade. Trade disputes may result in elevated tariffs and greater trade barriers, which could have a negative impact on China's import and export business [12–14].

In addition to the aforementioned factors, commodity quality and technology also have a great impact on China's import and export trade [15]. In the current competitive global market, it is crucial for China to enhance its product quality and technological level in order to improve its overall competitiveness [16]. Moreover, the significance of financial development in relation to China's import and export trade should not be underestimated [17]. The stability and sound development of the financial system are vital for trade financing and risk management [18]. Additionally, trade liberalization plays a significant role in China's import and export trade [19]. By reducing trade barriers and expanding market access, trade liberalization promotes the development of China's import and export trade [20].

Existing researches has predominantly utilized econometric methods, including gravity models, spatial econometric models, and panel data models, in analyzing spatio-temporal divergence. The selection of research methods is contingent upon the particular study's research topics and data availability. However, existing research are limited in the following ways: (1) Some studies focus solely on the global or international level of analysis, neglecting details at the provincial or regional level. (2) Some studies only implement a single econometric method for analysis, lacking multi-perspective comparison and validation.

The study contributes to the body of knowledge, influences public policy and has an economic and commercial impact. By thoroughly analysing the spatial and temporal heterogeneity of China's import and export trade and the various factors affecting its development, the study helps to better understand the internal mechanism and dynamic changes of international trade. This not only enriches the theory of international trade, but also provides us with a more comprehensive perspective to observe and explain the phenomenon of international trade. By revealing the spatial and temporal heterogeneity of China's import and export trade and the factors influencing it, companies can better understand market changes and predict future trends. This helps companies formulate more scientific and effective trade strategies and improve market competitiveness. Meanwhile, the study sheds light on developing countries' trade, helping them to better understand and exploit international trade opportunities, optimise resource allocation and promote economic development. At the same time, it provides the international community with a new way of thinking about cooperation that will help promote balanced and sustainable development of the global economy.

The following section presents our Theoretical analysis, which is followed by Research Methodology and Data Sources, Results of the study. The final section is the Conclusions and

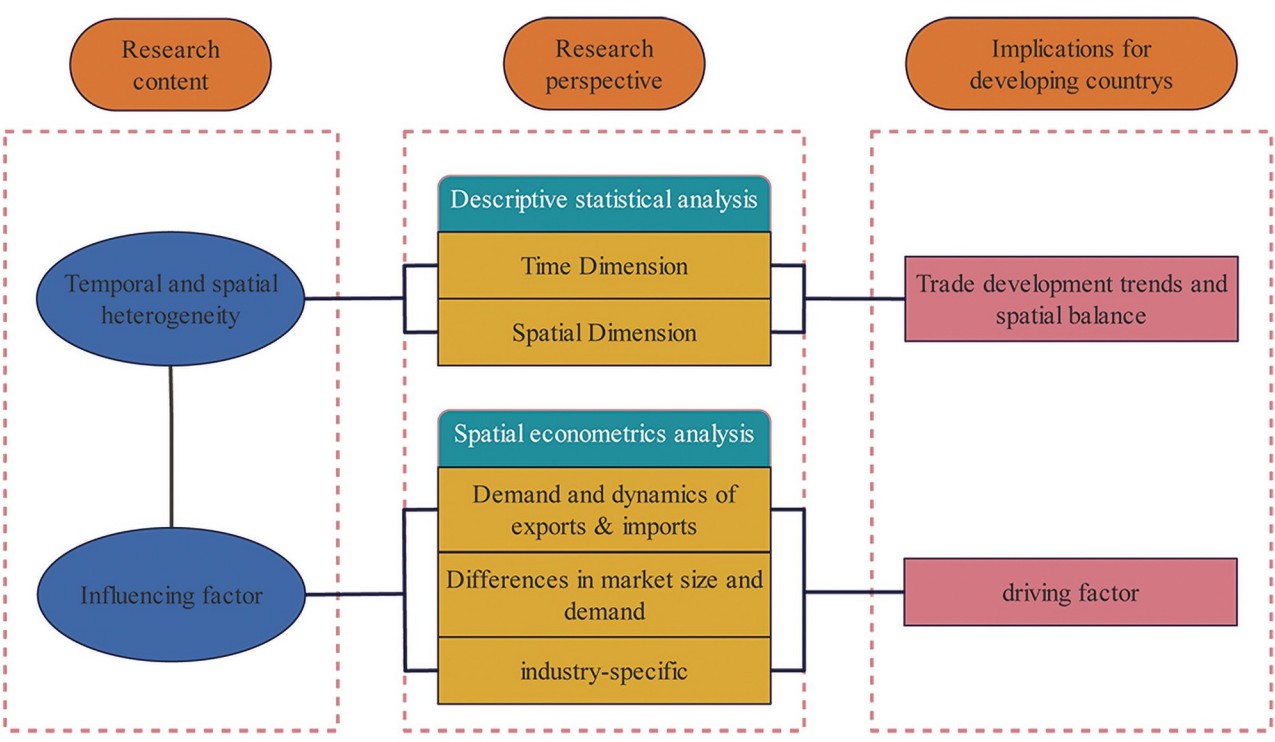

**Fig 1. Technical route.** This figure illustrates the research path of this article.

recommendations, divided into two subsections, the first subsection introduces Conclusions and recommendations of the study, the second subsection conducts Implications for developing countries. (Fig 1) illustrates the research path of this article.

## Theoretical analysis

### Core and periphery theory

Based on this theory, if in a region's product production there are scale economies and transportation costs such as the premise, there is a "∼" type three-curve relationship between the region's distance to economic center and its market potential, that is, increase in the distance of the region to economic center, will lead to the region's market potential first decline and then increase, but when the distance reaches a certain threshold value, the market potential will begin to decline with the increase in distance [21, 22]. At present, China's "East—Central —West" regional economic development is imbalance [23]. According to the "center—periphery" theory, regional big cities have a radiation driving effect on the development of the surrounding areas, by promoting the economic development of the West and other inland areas, and can improve the production capacity and resource supply capacity for foreign trade. According to the theory of "center-periphery", regional big cities have a radiation-driven effect on the development of the surrounding areas by promoting the economic development of the West and other inland areas, it can also improve the production capacity and resource supply capacity of the western region, and provide more products and resources for foreign trade exports. At the same time, this will also help to strengthen consumer demand in the inland market, providing a potential market for China's imports. In addition, realizing the

coordinated development of import and export trade in various regions can also improve China's international competitiveness and its position in the global value chain. At present, the global trade environment is facing uncertainties and great challenges, and international competition is becoming increasingly fierce. The coordinated development of import and export trade internally and externally can optimize China's industrial structure and the layout of the value chain [24], improve the quality and added value of the products, which is conducive to improving the competitiveness of Chinese enterprises in the international market, further consolidating and expanding China's position in the world economy, as well as providing reference for developing countries.

### Economies of scale theory

Economies of scale utilize production efficiencies to lower the average cost per unit of product [25]. China's labor resources enable to reduce cost per unit through scale effects, creating a competitive edge in manufacturing. China can achieve cost savings and offer these products at competitive international prices by concentrating on labor-intensive processing manufacturing and leveraging economies of scale. In recent years, China has redirected its export attention to technology-intensive and knowledge-intensive commodities from traditional labor-intensive ones. This change indicates China's endeavor to improve its industrial procedures to meet the alterations in global market demand, leading to higher value-added and augmented market competitiveness of its products.

### Classical trade theory

The English classical economist Adam Smith put forth the proposal in his 1776 volume "An Inquiry into the Nature and Causes of National Wealth" that the reason for international trade is the absolute cost disparity between countries [26]. If a country has an absolute advantage in producing a commodity at lower costs than other countries, then it can export that commodity; otherwise, it will import it. Due to its lower labor costs and abundant human resources, China has an advantage in human resources and labor expenses, enabling it to manufacture vast quantities of commodities at a reasonable price in the manufacturing industry, thereby acquiring a competitive edge in the global market. David Ricardo's Principles of Political Economy and Taxation, published in 1817, introduced the Law of Comparative Advantage. The theory suggests that international trade is based on the relative difference in production technology, not the absolute difference. As a result, there is a difference in relative costs. Each country should focus on producing and exporting commodities with a comparative advantage, while importing those with a comparative disadvantage, following the principle of "taking the greater of the two advantages and the lesser of the two disadvantages". Additionally, the principle of "choosing the lesser of two evils" should be applied [27]. The spatial and temporal heterogeneity of China's import and export trade is changing with the rapid development of China's economy and industrial upgrading. China is transitioning from exporting traditional, labor-intensive, and low-value-added products to exporting high-tech and high-value-added products. This shift is attributed to China's continuous efforts in scientific and technological innovation [28], talent development, and industrial upgrading.

## Research methodology and data sources

Our paper uses quantitative research methods, including the Gini coefficient of location, exploratory spatial analysis methods, and multiple linear regression.

## Locational Gini coefficient

The Gini coefficient is a tool used to quantify income distribution within a given region or country. It has since been extended to analyze the uneven distribution of other factors. The Locational Gini coefficient (LGC) quantifies the disparity in the geospatial distribution of economic activities. By providing insights into discrepancies in trade flows between regions, LGC can help to develop a coordinated inter-regional development plan. The formula for LGC is fundamental:

$$Gini^s = \frac{1}{2n^2\mu} \sum_{i=1}^{n} \sum_{j=1}^{n} |x_i^s - x_j^s| \tag{1}$$

Where the number of samples is $n$, the sample mean is $\mu$, and $x_i^s$ and $x_j^s$ indicate the proportion of import or export s in $i$ and $j$ regions to the total import and export trade of the country. To conveniently calculate the geospatial distribution imbalance of import and export trade, the absolute locational Gini coefficient is used, as per the study of Wu Aizhi (2013) [29]. The calculation formula for this coefficient is:

$$Gini^s = \frac{1}{2(n-1)} \sum_{i=1}^{n} \sum_{j=1}^{n} |x_i^s - x_j^s| \tag{2}$$

The Location Gini coefficient ranges from 0 to 1.

## Exploring spatial data analysis

Exploratory spatial data analysis is a technique utilized to research the distribution and spatial correlation of geographic occurrences. The spatio-temporal heterogeneity regarding China's import and export trade, which is influenced by various factors such as geographical location, regional economic development level, foreign investment, and market demand for consumption, produces divergent effects on import and export trade. Exploratory analysis of spatial data delves into the spatial laws and correlations behind import and export trade data through visual display and statistical analysis, which is crucial to uncovering and explaining the corresponding laws, patterns, and trends. This method plays a significant role in revealing spatial relationships. Spatial correlation indicators are categorized into global spatial autocorrelation indicators and local spatial autocorrelation indicators. Moran index [30] is used to test for global spatial autocorrelation indicators, while local indicators such as LISA cluster plot and Moran scatter plot are used to test for local spatial autocorrelation indicators. The calculation of the global is as follows:

$$Morans\ I = \frac{\sum_{i=1}^{n} \Sigma_{j=1}^{n} w_{ij}(x_i - \bar{x})(x_j - \bar{x})}{S^2 \sum_{i=1}^{n} \Sigma_{j=1}^{n} w_{ij}} \tag{3}$$

Where $S^2 = \frac{1}{n}\Sigma_{i=1}^{n}(x_i - \bar{x})^2$, $\bar{x} = \frac{1}{n}\sum_{i=1}^{n} x_i$. $x_i$ and $x_j$ represent observations in regions $i$ and $j$. $n$ denotes the total number of regions, while $w_{ij}$ represents the element at the intersection of the $i$ row and $j$ column of the spatial weights matrix. $w_{ij}$ utilizes a basic binary adjacency matrix.

Local $Morans'I_i$ is defined as

$$Morans'I_i = \frac{(x_i - \bar{x})}{S^2} \sum_{j=1}^{n} w_{ij}\left(x_j - \bar{x}\right) \tag{4}$$

A positive value for $Morans'I_i$ indicates that the spatial cell shares similar attributes with its neighboring cells ("high-high" or "low-low"), while a negative value for $Morans'I_i$ indicates that the spatial cell doesn't share similar attributes with its neighboring cells ("high-low" or "low-high").

## Multiple linear regression

Multiple linear regression is a statistical tool utilized to establish a linear association between several independent variables and a dependent variable. The regression coefficients are estimated through the method of least squares, which seeks to determine the combination of coefficients that minimizes the difference between the predicted and actual observed values. Using multiple linear regression analysis, we can ascertain the extent of contribution by various factors on the spatio-temporal heterogeneity of import and export trade. We can also identify the positive and negative relationships between independent and dependent variables. Furthermore, multiple linear regression analysis proves an effective tool in discerning future spatio-temporal heterogeneity of import and export trade trends. This provides policymakers the necessary reference basis for informed decision- making. Hence, multiple linear regression analysis emerges as among the more reasonable methodologies to analyze the spatio-temporal heterogeneity of China's import and export trade and the factors influencing it. The following is the multiple linear regression model expressed:

$$Y = \beta_0 + \beta_1 X_1 + \cdots + \beta_n X_n + \varepsilon \tag{5}$$

Where $Y$ represents the dependent variable's value, $X_1, X_2, \ldots, X_n$ stand for the independent variables' values, $\beta_0, \beta_1, \beta_2, \ldots, \beta_n$ denote the regression coefficients, and $\varepsilon$ refers to the error term.

Considering the data availability, the sample interval of the multiple linear regression is selected as panel data of 31 provinces (excluding Hong Kong, Macao, and Taiwan) from 2010 to 2021. The dependent variables included per capita exports and imports, while the independent variables comprised of GDP, fiscal expenditure, freight volume, technology market turnover, the rate of cell phone penetration, foreign investment, and disposable income of all residents.

## Data sources

The data utilized in this paper derive from the Department of Statistical Analysis of the General Administration of Customs of the People's Republic of China (http://stats.customs.gov.cn/) and the National Bureau of Statistics of China (http://www.stats.gov.cn/), covering the period from 2000 to 2022. It is worth mentioning that, due to data constraints, the time span for multiple linear regression is from 2010 to 2021.

## Results of the study

### Overview of spatial and temporal variation in China's import and export trade

Observing the total import and export commodities of 31 provinces, municipalities, and autonomous regions in China's interior from 2000 to 2022 (refer to Fig 2), significant differences are apparent in both temporal and spatial dimensions. Since 2000, the total import and export trade volume of China's provinces has shown an upward trend, mainly due to the following reasons: China's active participation in international economic cooperation and competition and its continuous expansion of the international market have provided a wider space

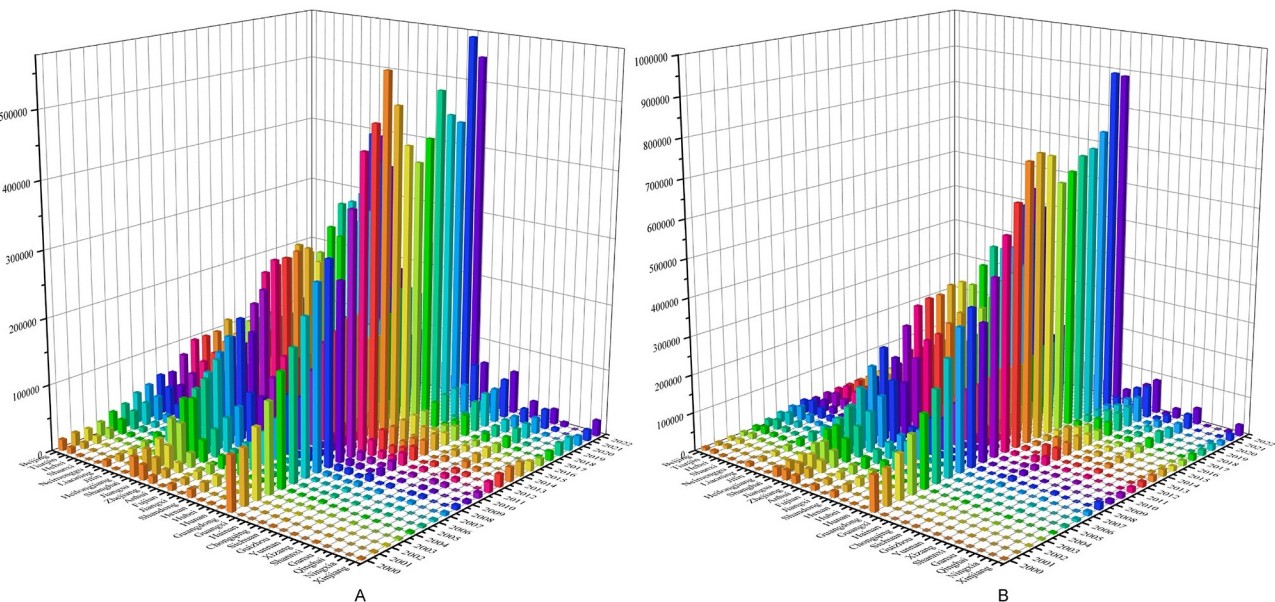

**Fig 2. Total import and export commodities by province in China, 2000–2022 (in millions of dollars).** A shows the statistical changes in China's import volume by province, 2000–2022;B shows the statistical changes in China's export value volume by province, 2000–2022. Source: China National Bureau of Statistics.

for the development of provincial import and export trade. At the same time, the government has introduced a series of policies to promote the development of foreign trade, such as export tax rebate, export credit, export insurance, etc., to encourage enterprises to expand their international market and improve their export competitiveness. Meanwhile, local governments have also actively introduced relevant policies to support local enterprises in import and export trade. Together with China's continuous strengthening of infrastructure construction and improvement of safety capacity in the fields of transportation, communication and energy, this has provided more convenient and efficient logistic support for import and export trade in various provinces.

(Fig 3) illustrates the spatial distribution of China's per capita import trade in 2000 and 2022. In 2000, China's per capita import trade demonstrated a Eastern Regional of China > Center Regional of China > Western Regional of China. This is attributed to the fact that the eastern coastal regions possess relatively developed economies, advanced ports and infrastructure, and a more established foreign trade environment. As the primary hubs for China's import and export trade, these areas have the capability to draw in additional foreign investment and international trade. In contrast, the average import trade per person in the central and western regions is lower, primarily because of the underdeveloped economy in these regions and the inadequate infrastructure and foreign trade support system.

In 2022, there are significant changes in the geographic distribution of China's per capita import trade. The eastern coastal region continues to exhibit the highest per capita import trade value in China. Although, the map demonstrates a gradual reduction in the disparity of the per capita import trade value through the darker and lighter color distribution throughout the country, the per capita value of imported commodities in the central and western regions has notably increased, particularly in the west. This suggests significant advancements in trade development within those areas, which are now playing a more substantial role in driving China's overall import trade growth.

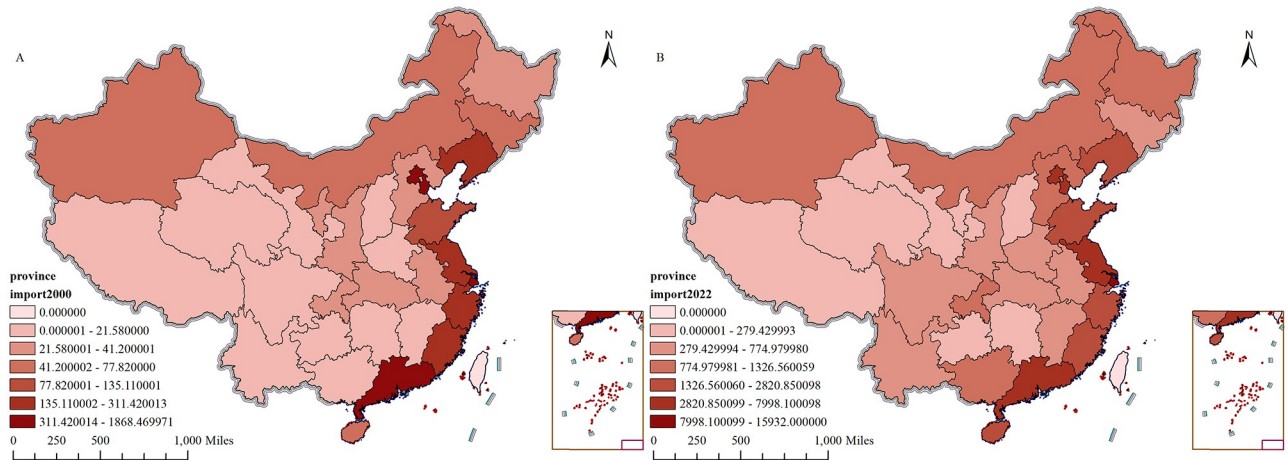

**Fig 3. China's import trade per capita, 2000, 2022 (in dollars).** A shows China's import trade per capita in 2000;B shows China's import trade per capita in 2022. The map is obtained from Natural Earth (http://www.naturalearthdata.com/).

Several factors account for this shift in geographic distribution. Initially, the Chinesegovernment has implemented a range of policies aimed at promoting development and exposing the central and western regions. Specifically, the "Western Development" policy has enhanced infrastructure construction and improved trade environments in these regions, thereby attracting more foreign investment and imports. Secondly, China is participating in the globalization process by promoting the Belt and Road Initiative and bolstering economic and trade collaboration with neighboring and Eurasian nations. This has created new possibilities and momentum for trade cooperation between central and western China and other countries.

In addition, the focus on economic restructuring and industrial upgrading in China has affected how import trade is distributed per capita. In recent decades, China's economy has gradually shifted from being export-oriented to driven by domestic demand, leading to an increase in demand for imported commodities among Chinese consumers. Due to China's rapidly growing economy and rising income levels among its people, the country's consumption patterns have evolved with an increase in demand for high-quality and specialty imported commodities. This has resulted in a further increase in China's per capita import trade volume.

(Fig 4) shows the spatial distribution of China's export trade per capita in 2000 and 2022. As can be seen from Fig 4, the spatial distribution of China's per capita export trade in 2000 and 2022 is still characterized by Eastern Regional of China > Center Regional of China > Western Regional of China. The reasons for this are the same as those explained in Fig 3 and will not be repeated here. The previous analysis suggests that China's per capita import and export trade has undergone substantial geographical alterations. Even though the eastern coastal area remains the central hub for import and export trade, the central and western regions have witnessed considerable improvement in per capita import and export trade, leading to a gradual reduction in the gap between per capita import and export trade. The shifting geographical distribution of China's per capita import and export trade suggests a growing role for the central and western regions in its development. This change is due to government policy support, improvements in infrastructure construction, and efforts to increase China's global participation. Despite progress, regional disparities persist and further policy measures are necessary.

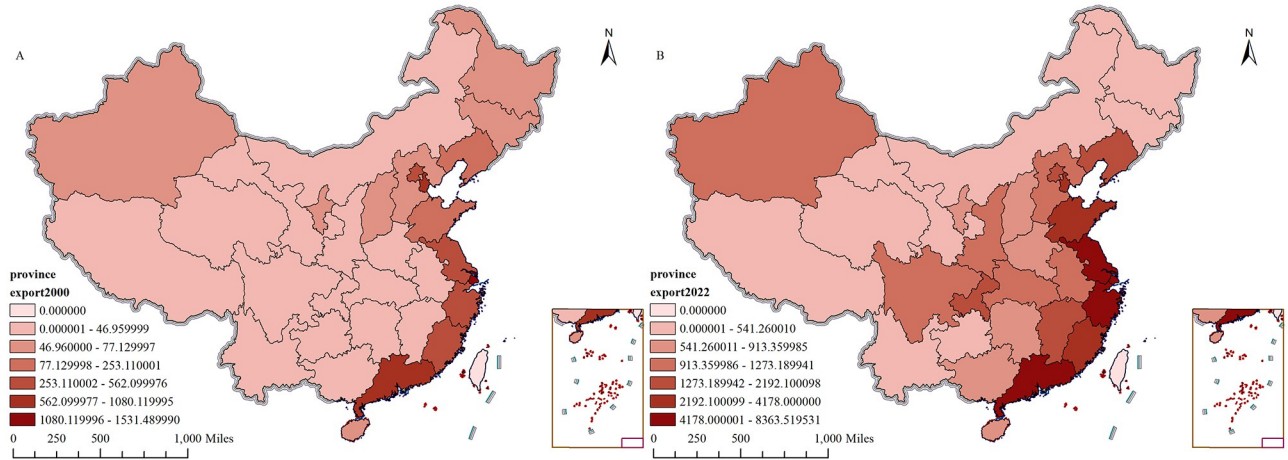

**Fig 4. China's export trade per capita, 2000, 2022 (in dollars).** A shows China's export trade per capita in 2000;B shows China's export trade per capita in 2022. The map is obtained from Natural Earth (http://www.naturalearthdata.com/).

## Locational Gini coefficient

By utilizing Eqs (1) and (2), we conducted an analysis of the Gini coefficient regarding China's import and export commodity locations (as illustrated in (Fig 5). The data displayed in Fig 5 clearly portrays a decline in the Gini coefficient, indicating a gradual balancing of spatial distribution for China's import and export commodities. The primary reason for this is that China is constantly promoting the development of trade liberalization and globalization, constantly optimizing the structure and layout of inter-regional trade, and reducing the imbalance in the flow of commodities between different regions.

In the period of 2000 and 2010, China's Locational Gini coefficient for imported and exported commodities decreased at a slower pace, likely due to the unbalanced economic development and trade policies of certain regions. Nevertheless, since 2010, China has introduced a set of opening-up policies, actively advocating for trade liberalization and facilitation, as well as strengthening trade cooperation with external regions. Driven by policies and measures, the regional Gini coefficient for China's import and export commodities has declined swiftly, indicating a more balanced distribution of regional trade.

Furthermore, the Gini coefficient of imported and exported commodities is impacted by variations in the levels of economic development and infrastructure construction across regions. Coastal areas and some inland provinces with advanced economic development have a competitive edge in trade, which affects the flow and distribution of imported and exported commodities to a certain degree.

## Moran index

From 2000 to 2022, China's per capita import and export global Moran index has undergone significant changes and developments, as presented in Table 1. Specifically, the index shows a clear upward trend, highlighting the strengthening trade connections among diverse regions within China. Further, these findings indicate a high correlation between the level of economic growth and trade development. This trend is primarily attributable to China's growing economy and expanding import and export trade.

In addition, China's per capita import and export global Moran index has exhibited some instability over time. From 2000 to 2008, China's per capita import and export global Moran

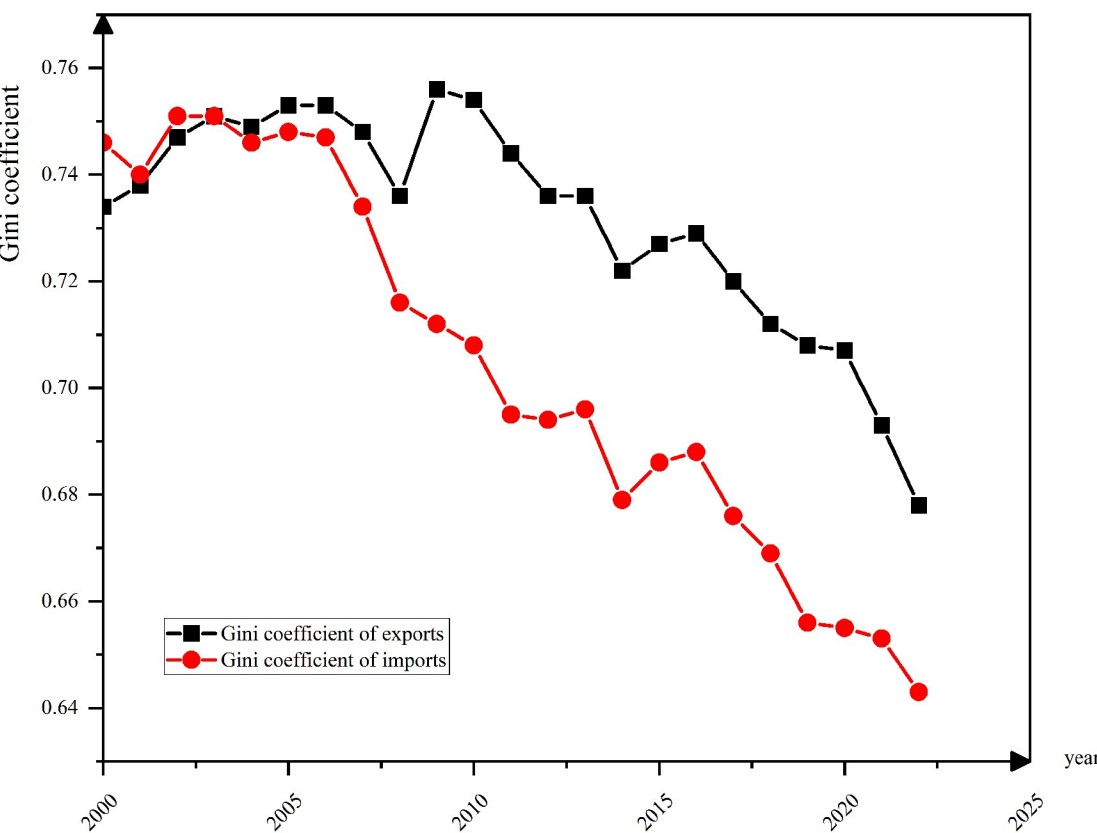

**Fig 5. Locational Gini coefficient of imported and exported commodities in China, 2000–2022.** The black line shows locational Gini coefficient of exported commodities in China, 2000–2022;The red line shows locational Gini coefficient of imported commodities in China, 2000–2022. Source: Department of Statistical Analysis, General Administration of Customs, People's Republic of China.

index witnessed a relatively stable upward trend. During this period, China's economy maintained a high growth rate, with an expanding import and export trade. Nevertheless, impacted by the global financial crisis in 2008, China's per capita import and export global Moran index experienced a certain decline. Subsequently, China's economy experienced a gradual recovery due to the government's proactive stimulus policies and internal and external demand. As a result, the per capita import and export global Moran Index rose once again. Since 2012, China's per capita import and export global Moran index has continued to grow, reaching a peak in 2020.

During this phase, growth is primarily attributable to the transformation and upgrade of the Chinese economy, as well as the structural adjustment of import and export trade. As China's manufacturing sector experiences rapid expansion, its import and export trade has transitioned from traditional labor-intensive products to technology-intensive and high-value-added products. The correlation between China's trade links and economic development with other countries has been further strengthened. However, after 2020, China's per capita global Moran index for import/export declined somewhat due to the impact of the COVID-19 epidemic. The epidemic resulted in a global economic slowdown and supply chain disruption, which had an impact on China's import and export trade as well as its economic development. However, as the epidemic is gradually brought under control and international trade begins to recover, China's per capita global Moran Index for imports and exports has started to rebound.

**Table 1. Global Moran index of China's per capita imports and exports, 2000–2022.**

| Variables | Export | | Import | |
|---|---|---|---|---|
| | Global Moran index | p-value* | Global Moran index | p-value* |
| 2000 | 0.2050** | 0.0240 | 0.1860** | 0.0350 |
| 2001 | 0.2280** | 0.0140 | 0.1950** | 0.0270 |
| 2002 | 0.2390** | 0.0120 | 0.1960** | 0.0240 |
| 2003 | 0.2480*** | 0.0080 | 0.1970** | 0.0150 |
| 2004 | 0.2610*** | 0.0040 | 0.2320*** | 0.0060 |
| 2005 | 0.2890*** | 0.0020 | 0.2550*** | 0.0040 |
| 2006 | 0.3060*** | 0.0010 | 0.2730*** | 0.0030 |
| 2007 | 0.3210*** | 0.0010 | 0.2840*** | 0.0020 |
| 2008 | 0.3410*** | 0.0000 | 0.3110*** | 0.0010 |
| 2009 | 0.3650*** | 0.0000 | 0.3080*** | 0.0010 |
| 2010 | 0.3860*** | 0.0000 | 0.3040*** | 0.0010 |
| 2011 | 0.3760*** | 0.0000 | 0.3100*** | 0.0010 |
| 2012 | 0.3530*** | 0.0000 | 0.2890*** | 0.0010 |
| 2013 | 0.3300*** | 0.0010 | 0.2760*** | 0.0030 |
| 2014 | 0.3370*** | 0.0010 | 0.2750*** | 0.0030 |
| 2015 | 0.3660*** | 0.0000 | 0.2540*** | 0.0030 |
| 2016 | 0.3860*** | 0.0000 | 0.2380*** | 0.0050 |
| 2017 | 0.4030*** | 0.0000 | 0.2450*** | 0.0030 |
| 2018 | 0.4180*** | 0.0000 | 0.2470*** | 0.0040 |
| 2019 | 0.4230*** | 0.0000 | 0.2260*** | 0.0060 |
| 2020 | 0.4190*** | 0.0000 | 0.2300*** | 0.0030 |
| 2021 | 0.4380*** | 0.0000 | 0.2330*** | 0.0020 |
| 2022 | 0.4840*** | 0.0000 | 0.2480*** | 0.0020 |

Note:

***, **, * indicate significant at 1%, 5%, 10% level, respectively.

In summary, the per capita global Moran index for imports and exports in China demonstrates a rising trend, indicating trade connections between various regions of the country and an enhancement in economic development. While some fluctuations may occur due to external factors, overall, the trend reflects consistent growth. In the future, as China's economy continues to grow and the import and export trade expands, there is an expectation that the country's per capita import and export global Moran Index will increase further, resulting in improved trade links and increased economic development.

(Fig 6) shows the LISA (Local Indicators of Spatial Autocorrelation) agglomeration map of China's per capita imports in 2000 and 2022. In 2000, regions with higher per capita imports in China were close to each other, forming a spatial agglomeration patterns. The clustered regions are situated in the eastern coastal areas, specifically in Guangdong, Shanghai, and Zhejiang. These regions typically boast a more advanced economy, with accessible transportation, a varied industrial composition, and a relatively elevated import demand.

In 2022, the clustered regions in China with higher per capita imports will be further strengthened. Over the past two years, these regions have seen increased imports per capita due in part to the rapid development of the global economy and the further opening of the Chinese economy to the world. Additionally, new agglomeration areas have emerged. These new agglomerations may arise from several factors, including heightened levels of economic growth, policy reforms, or the establishment of new special economic zones.

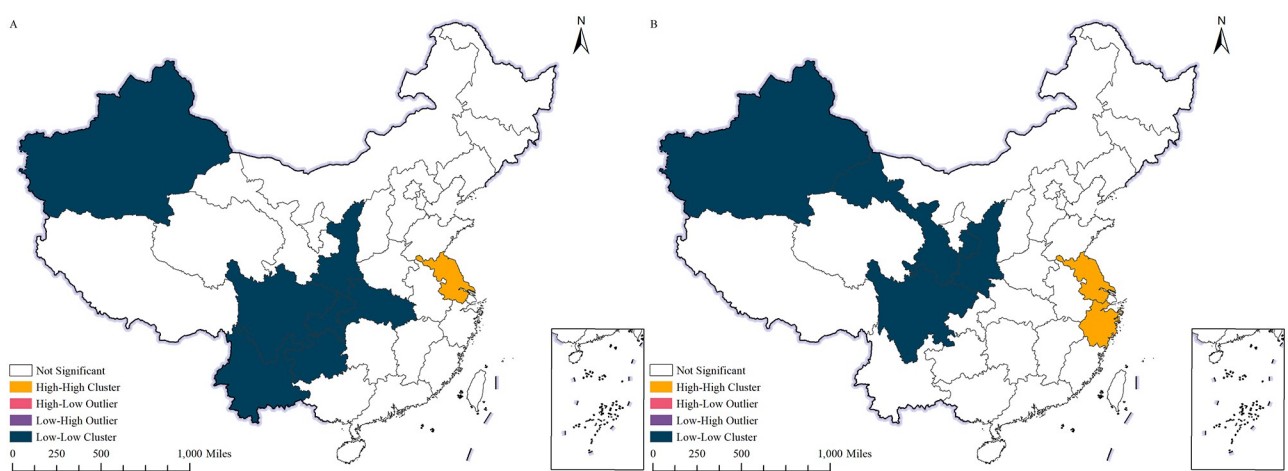

**Fig 6. LISA clustering of China's imports per capita, 2000, 2022.** A shows the LISA clustering of China's imports per capita, 2000;B shows the LISA clustering of China's imports per capita, 2022. The map is obtained from Natural Earth (http://www.naturalearthdata.com/).

The clustering characteristics of per capita imports are influenced by several factors. Among them, geographic location holds significant influence. Coastal and adjacent developed regions generally record higher per capita imports due to the favorable transport and logistics conditions which promote the growth of import trade. Secondly, economic development level represents a crucial factor that impacts the per capita import volume. Regions that are economically developed typically have a diversified industrial structure, leading to a high dependence on import demand. Furthermore,government policies and regulatory measures influence the agglomeration characteristics of imports per capita. The government's open immigration policy and preferential trade policies have the potential to increase the volume of imported commodities and stimulate the concentration of imports per person.

It is noteworthy that the presence of agglomeration does not necessarily entail dominance. Areas characterized by agglomeration and high per capita imports may suggest a greater degree of import dependence, while the absence of agglomeration in other regions may not necessarily point to economic underdevelopment, but rather reflect a diverse range of economic structural and policy-related factors. Consequently, a detailed and comprehensive analysis of relevant data, coupled with in-depth studies, is required to evaluate the true level of economic development and competitiveness of regions that exhibit agglomeration.

(Fig 7) displays the LISA (Local Indicators of Spatial Autocorrelation) agglomeration map illustrating China's per capita export value in 2000 and 2022. Spatial distribution of China's per capita export value still exhibits obvious agglomeration. Notably, the eastern coastal region features significantly higher per capita export value than other regions, specifically Guangdong and Zhejiang. Conversely, per capita export value in the central and western regions is comparatively lower.

The comparison between 2000 and 2022 reveals an expansion in the scope of agglomeration. This expansion can be attributed to China's increasing openness to the outside world, which has accelerated economic growth in the central and western regions. Additionally, the scale of export trade has expanded, resulting in an increase in per capita export value. Furthermore, it is evident that the per capita export value of certain inland regions, including Chongqing and Sichuan, has risen. This is primarily attributed to the accelerated economic growth within these regions in recent years, resulting in an expansion of export trade.

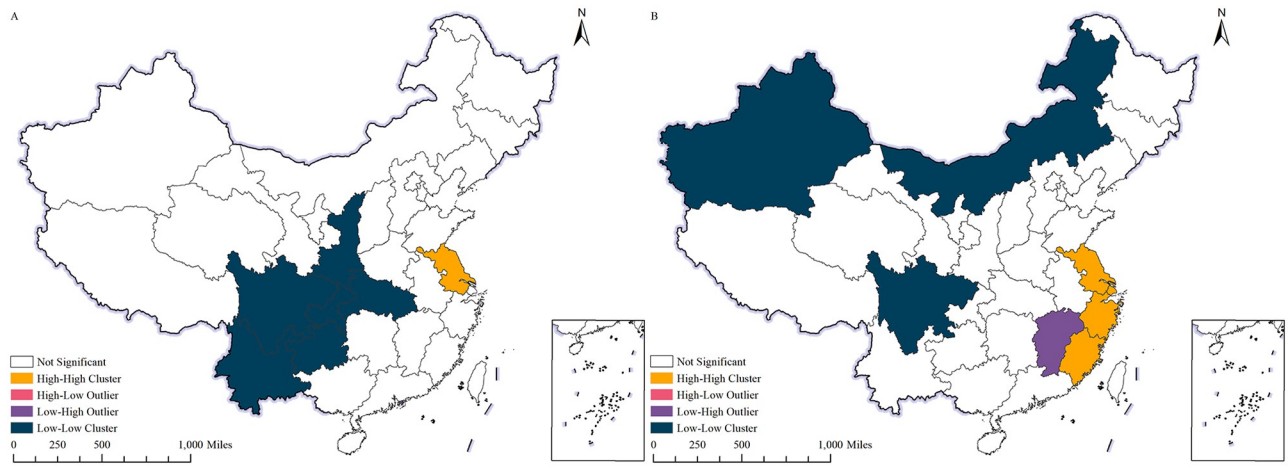

**Fig 7. LISA clustering of China's exports per capita, 2000, 2022.** A shows the LISA clustering of China's exports per capita, 2000;B shows the LISA clustering of China's exports per capita, 2022. The map is obtained from Natural Earth (http://www.naturalearthdata.com/).

Overall, the spatial distribution of China's exports per capita has gradually become more uniform from 2000 to 2022. This indicates an improvement in the balance of China's economic development, and a reduction in the disparity between the economic development levels and openness of different regions. Meanwhile, it indicates that China's approach to opening its economy is transitioning from coastal to inland areas and all regions are actively engaged in expanding their global reach to facilitate China's economic advancement.

## Multiple linear regression results

Multiple linear regression analysis was conducted using Eviews10, and Tables 2 and 3 display the results. The data indicates that the six independent variables, namely GDP, financial

**Table 2. Results of the analysis of factors affecting the value of exports per capita.**

| Variable | Coefficient | Std. Error | t-Statistic | Prob. |
|---|---|---|---|---|
| C | -2732.1780*** | 279.8754 | -9.7621 | 0.0000 |
| ?GDP | 0.0273*** | 0.0049 | 5.5969 | 0.0000 |
| ?FINANCE | 0.2796*** | 0.0204 | 13.6954 | 0.0000 |
| ?HUOYUN | -0.0053*** | 0.0008 | -6.4156 | 0.0000 |
| ?JISHU | -1.0099*** | 0.0818 | -12.3448 | 0.0000 |
| ?TEL | 0.3887 | 3.6837 | 0.1055 | 0.9160 |
| ?INVEST | 0.0007*** | 0.0002 | 4.2151 | 0.0000 |
| ?INCOME | 0.1308*** | 0.0094 | 13.9901 | 0.0000 |
| R-squared | 0.7588 | Mean dependent var | | 1332.5530 |
| Adjusted R-squared | 0.7541 | S.D. dependent var | | 1881.0580 |
| S.E. of regression | 932.7142 | Akaike info criterion | | 16.5354 |
| Sum squared resid | 317000000.0000 | Schwarz criterion | | 16.6196 |
| Log likelihood | -3067.5740 | Hannan-Quinn criter. | | 16.5688 |
| F-statistic | 163.5675 | Durbin-Watson stat | | 0.1532 |
| Prob(F-statistic) | 0.0000 | | | |

Note:

***, **, * indicate significant at 1%, 5%, 10% level, respectively.

**Table 3. Results of the analysis of factors affecting the value of imports per capita.**

| Variable | Coefficient | Std. Error | t-Statistic | Prob. |
|---|---|---|---|---|
| C | -2538.2060*** | 353.7439 | -7.1753 | 0.0000 |
| ?GDP | -0.0277*** | 0.0062 | -4.4987 | 0.0000 |
| ?FINANCE | 0.3838*** | 0.0258 | 14.8753 | 0.0000 |
| ?HUOYUN | -0.0058*** | 0.0011 | -5.5518 | 0.0000 |
| ?JISHU | -0.6183*** | 0.1034 | -5.9795 | 0.0000 |
| ?TEL | -16.2375*** | 4.6559 | -3.4875 | 0.0005 |
| ?INVEST | 0.0010 *** | 0.0002 | 4.9793 | 0.0000 |
| ?INCOME | 0.2234*** | 0.0118 | 18.9007 | 0.0000 |
| R-squared | 0.7406 | Mean dependent var | | 1402.3890 |
| Adjusted R-squared | 0.7357 | S.D. dependent var | | 2292.9000 |
| S.E. of regression | 1178.8890 | Akaike info criterion | | 17.0038 |
| Sum squared resid | 506000000.0000 | Schwarz criterion | | 17.0881 |
| Log likelihood | -3154.7070 | Hannan-Quinn criter. | | 17.0373 |
| F-statistic | 148.4936 | Durbin-Watson stat | | 0.1636 |
| Prob(F-statistic) | 0.0000 | | | |

Note:

\*\*\*, \*\*, \* indicate significant at 1%, 5%, 10% level, respectively.

expenditure, freight transportation, technology market turnover, foreign investment, and disposable income of all residents, have a statistically significant impact on both the per capita amount of exports and imports. The penetration rate of cell phone usage has a notable impact only on the per capita amount of imports, while it does not have a significant effect on the per capita amount of exports.

## Discussion

### The unique aspects and contributions of this study

The results of this study have numerous important explanations and implications. Firstly, through the in-depth study of the causes and influence mechanisms of spatio-temporal heterogeneity, it can provide new perspectives and methods for understanding the laws of trade development in depth, and provide theoretical support for the formulation and adjustment of trade policies. Secondly, this study examines the spatio-temporal variations in international trade and its contributing factors using provincial panel data. This analysis can assist future research in developing a more profound comprehension of the distribution patterns of international trade and its influencing factors at the provincial and regional levels. Thirdly, understanding the characteristics and influencing factors of the spatial and temporal heterogeneity of import and export trade can assist the government in enhancing its trade strategy, optimizing its trade structure, improving trade efficiency, carrying out diversified trade activities and market diversification, developing a scientifically formulated trade policy, strengthening interregional economic cooperation, and promoting balanced trade development. Finally, this paper provides significant inspiration and guidance for the trade development of other developing countries. To cope with challenges and obstacles in international trade, developing countries should improve cooperation with other countries and international organizations. Strengthening cooperation increases access to technology, capital, and market resources, which ultimately improves international competitiveness and trade levels.

### Limitations of this study

This study has limitations. Due to data availability and processing difficulties, only select indicators were used to measure import/export trade in various regions of China. In the future, more comprehensive data can provide a more accurate portrayal of the import/export trade situation. Secondly, this study only analyzes a select few major factors affecting import and export trade, other potential influencing factors need to be further studied in the future. Furthermore, while this study delves extensively into the functioning mechanisms of each of these influencing factors, additional empirical research is necessary to confirm their actual impact.

## Conclusions and recommendations

### Conclusions and recommendations of the study

The study in this paper concludes that there are significant spatial and temporal differences in China's import and export trade, which are influenced by factors such as GDP, fiscal expenditure, freight transport, technology market turnover, foreign investment and residents' disposable income. The results of the study show that these factors have a significant impact on per capita import and export volumes. The study also found that from 2000 to 2022, China's export trade centre will gradually shift to inland regions, and the regions will actively expand their foreign trade, which will promote the development of China's economy. The results based on multiple linear regression analyses show that six independent variables, including GDP, fiscal expenditure, freight volume, technology market turnover, foreign investment and residents' disposable income, have a significant impact on the amount of per capita imports and exports. The penetration rate of mobile phone use only has a significant effect on the amount of imports per capita, and has no significant effect on the amount of exports per capita. Therefore, it is recommended that China and other developing countries take effective measures to promote balanced trade development, strengthen regional cooperation and coordination, and promote green trade and sustainable development. The specific findings are as follows:

(1) From 2000 to 2022, the per capita import and export values exhibited varying degrees of growth trends for each province. However, inter-provincial import and export trade displayed distinct spatio-temporal differentiation characteristics with a prominent east-to-west disparity.

(2) Due to the influence of Chinese government policies, there is a declining Gini coefficient for China's import and export locations, resulting in a more balanced spatial distribution of these commodities.

(3) From 2000 to 2022, China's per capita import and export trade demonstrated an overall upward trend in fluctuation, as evidenced by the global Moran index. In addition, the trade links between various regions of the country have gradually strengthened, with a corresponding high correlation to the level of economic development.

(4) The spatial distribution of China's foreign trade has resulted in an agglomeration pattern focused in the eastern coastal region. Meanwhile, economic development has been accelerating in the central and western regions, leading to a more balanced regional development overall.

(5) The six independent variables, including GDP, financial expenditure, freight transportation volume, technology market turnover, foreign investment, and disposable income of all

residents, significantly influence the per capita export and import volume. To ensure sustainable growth of China's import and export trade and regional balance, these factors can be targeted.

The following recommendations are made accordingly:

(1) Prioritizing the development of regions with competitive advantages. Based on spatial and temporal distribution characteristics and trade advantages, priority should be given to selecting and developing regions with competitive advantages. To this end, efforts should be made to optimize the industrial structure, increase technological innovation and the development of high value-added industries, improve product quality and added value, and enhance competitiveness. On the other hand, the government should enhance transportation infrastructure construction in the central and western regions and bolster regional connectivity to enhance their connection with external markets. This could facilitate the growth of import and export trade in the central and western regions and improve their competitiveness.

(2) Strengthening regional strategies for openness and promoting industrial structure upgrades. Regions with clear geographic advantages, such as coastal cities and border areas, need to strengthen their connections to the larger world and create a more open economic system, ultimately increasing the per capita import and export value of the region. To achieve these goals, measures such as expanding exports and attracting foreign investment will be necessary. Guiding the internationalization of businesses, Increasing support for exporting by small and medium-sized enterprises, promoting overseas market expansion, establishing overseas sales networks and production bases, bolstering international competitiveness, and elevating the region's per capita import and export value. By implementing policies and strategies, leveraging geographic and industrial advantages, boosting the growth of distinctive industries and favorable products, enhancing trade conditions, improving workforce skills, and increasing international partnerships, the per capita import and export value of each region can be significantly increased.

(3) Encouraging regions to bolster trade relationships and promote collaboration among companies across regions by organizing trade shows, exchanges, and other activities to achieve complementarity of advantages. Increase in investment in transportation, communication, and infrastructure is need to enhance trade efficiency and facilitation, thus reducing trade expenses even further.

(4) Increasing investment and support for the central and western regions to enhance economic development and competitiveness. It is necessary to strengthen infrastructure construction and talent training in these regions in order to raise their economic strength. And it is also necessary to promote coordinated regional development, facilitate economic exchanges and collaboration among the eastern, central, and western regions, and achieve optimal resource allocation.

(5) Sustaining China's import and export trade and regional balance, the following areas deserve attention: enhancing economic and trade competitiveness, streamlining fiscal expenditure structure, improving logistics and transportation infrastructure efficiency, promoting technological innovation and scientific achievements transformation, optimizing foreign investment environment, and elevating income levels of the population to bolster consumer demand.

### Implications for developing countries

(1) emphasizing balanced regional development. Developing countries must recognize the spatial and temporal characteristics of import/export trade and endeavor to promote a balanced regional development. Through strengthening economic ties and cooperation between regional areas, it is possible to promote a balanced distribution of trade and reduce inter-regional disparities.

(2) Optimizing the policy environment. Policies significantly impact trade development. Developing countries should adjust policies to reduce the geographical Gini coefficient of imports and exports and promote a balanced trade distribution. They should also establish a policy environment that fosters trade development by offering support and incentives, reducing trade barriers, and facilitating trade.

(3) Strengthening Economic Ties and Cooperation. The ongoing upward trend in the global Moran Index of per capita import and export trade indicates a gradual strengthening of trade links between various regions. Developing countries must actively engage in international trade cooperation, strengthen economic ties with other countries, and promote the diversification of trading partners in order to realize greater market access and trade opportunities.

(4) Promoting sustainable development. Developing countries need to prioritize sustainable development in their foreign trade development. In order to achieve this objective, they should strengthen the balance between environmental protection, social responsibility, and economic development. Furthermore, they should encourage the promotion of green trade and sustainable development practices, enhance the quality and competitiveness of their products, and foster long-term sustainable trade development.

## Supporting information

**S1 Table.**
(XLSX)

**S2 Table.**
(CSV)

**S3 Table.**
(CSV)

**S1 Data.**
(XLSX)

**S1 File.**
(PDF)

## Author Contributions

**Data curation:** Yuhuan Wu.

**Visualization:** Ning Zhu.

**Writing – original draft:** Haisong Wang.

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
