## [Decision Letter · Decision Letter 0]

30 Oct 2023

PONE-D-23-33006Analysis of the Characteristics of Spatial and Temporal Divergence of China's Import and Export TradePLOS ONE

Dear Dr. Wu,

Thank you for submitting your manuscript to PLOS ONE. After careful consideration, we feel that it has merit but does not fully meet PLOS ONE’s publication criteria as it currently stands. Therefore, we invite you to submit a revised version of the manuscript that addresses the points raised during the review process.

We look forward to receiving your revised manuscript.

Kind regards,

Rita Yi Man Li

Academic Editor

PLOS ONE

Journal Requirements:

5. Please amend the manuscript submission data (via Edit Submission) to include author  Zhu Ning.

6. We note that Figures 3, 4, 6 and 7 in your submission contain map/satellite images which may be copyrighted. All PLOS content is published under the Creative Commons Attribution License (CC BY 4.0), which means that the manuscript, images, and Supporting Information files will be freely available online, and any third party is permitted to access, download, copy, distribute, and use these materials in any way, even commercially, with proper attribution. For these reasons, we cannot publish previously copyrighted maps or satellite images created using proprietary data, such as Google software (Google Maps, Street View, and Earth). For more information, see our copyright guidelines: http://journals.plos.org/plosone/s/licenses-and-copyright.

a. You may seek permission from the original copyright holder of Figures 3, 4, 6 and 7 to publish the content specifically under the CC BY 4.0 license.  

Additional Editor Comments:

1. Originality:

• The paper lacks significant new information to justify publication. It mainly summarizes and analyzes existing literature without providing novel insights or findings.

2. Relationship to Literature:

• Many paragraphs have missing citations.

• Regarding technological innovation, citation and more details are needed: Is centralization killing innovation? The success story of technological innovation in fiscally decentralized countries, Technological Forecasting and Social Change Volume 168, July 2021, 120731

• The paper demonstrates a limited understanding of the relevant literature in the field. It relies heavily on a few sources and neglects to cite a broader range of literature that could provide a more comprehensive context for the research.

•

3. Methodology:

• The paper's argument is not adequately built on an appropriate base of theory or concepts. The theoretical framework is unclear, and it would benefit from a more robust foundation to support the research.

• The design of the research or intellectual work is not well-described. The methods employed are not clearly explained, and it is difficult to assess their appropriateness for the research objectives.

• Mutltiple regression needs citation: Comparative Study of Factors Contributing to Land Surface Temperature in High-Density Built Environments in Megacities Using Satellite Imagery, Sustainability 2021, 13(24), 13706

4. Results:

• The presentation of results lacks clarity and coherence. The data are not effectively analyzed or interpreted, making it challenging to draw meaningful conclusions from them.

• The conclusions do not adequately tie together the other elements of the paper. They appear disjointed and unsupported by the evidence presented.

5. Implications for research, practice, and/or society:

• The paper fails to clearly identify any implications for research, practice, and/or society. It does not bridge the gap between theory and practice, and there is a lack of discussion on how the research findings can be practically applied or have an impact on economic, commercial, or public policy aspects.

• The paper does not adequately discuss the potential contributions to the body of knowledge in the field. It does not position the research within the existing published work, and the referencing and introductory discussion are insufficient.

6. Quality of Communication:

• The paper's clarity of expression and readability need improvement. The sentence structure is often convoluted, and there is excessive use of technical jargon and acronyms that may hinder understanding for readers outside the specific field.

• The paper is not written concisely. There is unnecessary repetition of information and a lack of focus on the key points, making it challenging for readers to grasp the main ideas.

• The organization of the paper could be enhanced. The flow of information is disjointed in some sections, and the logical progression of the argument is not always clear.

• Some figures and tables lack clear explanations or citations to support the data presented. This compromises the overall clarity and reliability of the visual representations.

General Comments:

• The research question in the introduction is not clearly stated. It should be refined to provide a concise and focused overview of the study's objective.

• The introduction section exceeds the recommended length. It would be beneficial to shorten it by eliminating unnecessary background information and focusing on the specific research problem and objectives.

• There are several citation errors throughout the paper. Inconsistent referencing styles and missing or incorrect citations undermine the credibility and academic rigor of the work.

• The interpretations and conclusions drawn in the paper are not consistently supported by the evidence presented. There are instances where assumptions are made without sufficient justification, and the logical connection between the evidence and conclusions is weak.

• The title of the paper does not adequately represent the content. It should be revised to accurately reflect the scope and focus of the research.

• The abstract provides a brief summary of the paper but lacks clarity and conciseness. It should be revised to clearly state the main results and provide a more accessible overview of the study.

• The keywords chosen do not accurately reflect the content of the paper. They should be carefully selected to align with the main themes and concepts addressed in the research.

• The length of the paper is appropriate for the level of detail and analysis presented.

• The key messages conveyed in the paper should be shortened, ensuring they are accurate, clear, and succinct to effectively communicate the main findings.

• There are no concerns about plagiarism identified in the paper.

These comments aim to provide constructive feedback to improve the paper's quality, coherence, and impact.

Reviewers' comments:

Reviewer's Responses to Questions

**Comments to the Author**

1. Is the manuscript technically sound, and do the data support the conclusions?

Reviewer #1: Yes

Reviewer #2: Partly

Reviewer #3: Yes

Reviewer #4: Yes

Reviewer #5: Partly

2. Has the statistical analysis been performed appropriately and rigorously? 

Reviewer #1: Yes

Reviewer #2: No

Reviewer #3: Yes

Reviewer #4: Yes

Reviewer #5: Yes

3. Have the authors made all data underlying the findings in their manuscript fully available?

Reviewer #1: Yes

Reviewer #2: Yes

Reviewer #3: Yes

Reviewer #4: Yes

Reviewer #5: Yes

4. Is the manuscript presented in an intelligible fashion and written in standard English?

Reviewer #1: Yes

Reviewer #2: No

Reviewer #3: Yes

Reviewer #4: Yes

Reviewer #5: Yes

5. Review Comments to the Author

Reviewer #1: Dear Authors,

Thank You for so relevant and exciting Article.

Please consider the Sustainability concept concerning the Article topic. I'd recommend considering ESG-goal achievement in the Discussion section.

Reviewer #2: Your topic is very interesting Title: " Analysis of the Characteristics of Spatial and Temporal Divergence of China's Import and Export Trade ". I have some comments on your scripture which help you to update your manuscript.

Introduction

The introduction section doesn’t present clear picture what we need to need to learn, why we need to learn. Problem statement, research objectives or significance of the study should be clearly stated.

Literature Review

The literature review section lacks theoretical underpinnings. In my opinion the whole section needs to overhaul. The researcher need to read past relevant studies and also make sure that you attempt to identify the gaps in the existing literature such that it becomes easy to justify and trace the contribution of your study. Moreover, scant studies helps you to identify the appropriate theory on which your relationship will be based.

Methodology

The study has critical deficiencies from a methodological standpoint. How the researcher determined the sample size? Where is the justification? In the current study there is no reference (Few but not related) to any sort of information about generating an item pool and receiving expert opinions for the scale. Moreover, what are the sources of scales? How it was developed? Unfortunately, it is a doubt whether the scales are valid. No information exits about a validity test for scales. Furthermore, How to overcome the common method bias as data has been collected from single method ? where is CFA ? THERE IS NO THEORETICAL GROUNDING..

Discussion

The tests and presentation of findings are much more complicated. Again, I had difficulties in understanding the how findings of the study are connected to results of the analyses. Hypotheses testing is not well stated. The same applies to conclusion section

Contribution

The study's contribution is not written (a few of their but theoretical contribution is missing). Who will benefit from this research, both practically and theoretically? What role will it play in the existing literature? What's the latest/new?

Reviewer #3: The core claims are clearly expressed in abstract and conclusion part of the research paper. The empirical findings are cohere with each other and theorical framework of this work. The literature review is quite extensive and completed using up-to-date sources on the subject. The methods applied and the variables used in the methodology section of the article provided the required results. The findings of the article contributed to the literature as well. The findings and novelty of the manuscript contributes the existing literature. The English language of the article is clear and understandable, and the terminology used is compatible with the subject of the study. The article is very well-written and it is easy to follow. In addition, the article well discusses the topic of interest, and deals with a topic with many applications in practice. The quality of literature review and the aim of the paper is pretty well. The quality of the research is acceptable as well as the contribution of the paper. The quantitative research is very well conducted and the methodology described in the specific chapter is followed by current data processing. The methodology used is also very well described. The research fills a gap both by focusing on China's trade capacity and the influencing factors by considering the characteristics of spatial and temporal divergence to conduct exhaustive research on the influencing factors and temporal-spatial divergences in China's import and export trade. The researchers of the article recommend ensuring the optimization of trade channel layout. To enhance transportation efficiency, accessibility of goods, and decrease trade expenses, trade channel layout must be rationalized according to the spatial and temporal distribution traits of import and export trades. Firstly, upgrading of seaports and inland ports is essential for ameliorating goods' loading, unloading and transportation capacities, and accelerating trade transport times. On the other hand, diversifying and improving trade channels can be achieved through developing logistics networks, including highways and railroads. Additionally, facilitating faster and more convenient trade channels can be achieved by promoting the advancement of air freight and logistics technology. Strengthening trade cooperation and communication is also essential. This can be accomplished by actively expanding international trade partners and enhancing trade cooperation and communication with major trading partner countries and regions. Strengthening trade cooperation with developed economies is one approach to enhancing our trade level by learning advanced trade concepts and management experience. Additionally, our trade activities can be diversified and economic complementarity can be strengthened by focusing on trade cooperation with emerging market and developing countries. Furthermore, there is a need to establish and expand free trade zones and regional economic cooperation mechanisms to enhance trade facilitation and increase the scope of trade. Upgrading Trade Technology: Enhancing trade-related technological research and innovation to analyze and forecast trade data more effectively. Utilize innovative technologies, such as big data analysis and artificial intelligence, to gain deeper insights into trade potential and predict trade demand and market fluctuations accurately. Additionally, strengthen trade information sharing and platform development to ensure timely and precise trade information. Furthermore, the emphasis should be on developing trade talent by enhancing training and exchange programs, as well as improving the technical capabilities and professional quality of practitioners. In my opinion, the article has its merit and is of interest for the PLOS One readership. Consequently, the article is acceptable...

Reviewer #4: Your work contains multiple analyses, is remarkable as such, and is gripping to read. My suggestions and editing requests for the study were sent to the editor as a pdf. I wish you success in your future manuscripts.

Reviewer #5: Although the subject of the article is attractive, there are many issues that need to be corrected in the article.

1-) The purpose of the article is not fully described in the introduction part of the article. What is the purpose of the article? Why is the topic of the article important? What is the article's contribution to the literature?

2-) Gini coefficient needs more detailed explanation.

3-) ESA and MLR also need more explanation.

4-) Conclusion section is too short.

5-) Is there no discussion section? The authors need to add a discussion section in the article.

6. PLOS authors have the option to publish the peer review history of their article (what does this mean?). If published, this will include your full peer review and any attached files.

Reviewer #1: **Yes: **Sergey Barykin

Reviewer #2: **Yes: **RASHIDN MD SALAMUN

Reviewer #3: **Yes: **Salih Kalaycı

Reviewer #4: **Yes: **M. Esra Atukalp (Assoc. Prof. Dr.)

Reviewer #5: No

---

## [Author Response · Author response to Decision Letter 0]

28 Dec 2023

Reviewer #1

Dear Authors,

Thank You for so relevant and exciting Article.

Please consider the Sustainability concept concerning the Article topic. I'd recommend considering ESG goal achievement in the Discussion section.

The author’s response 1.1

We have revised the Conclusions and Recommendations section to include the ESG objective under Implications for Developing Countries, and have adjusted the topic accordingly.

Reviewer #2

Your topic is very interesting Title: " Analysis of the Characteristics of Spatial and Temporal Divergence of China's Import and Export Trade ". I have some comments on your scripture which help you to update your manuscript.

Introduction

The introduction section doesn’t present clear picture what we need to need to learn, why we need to learn. Problem statement, research objectives or significance of the study should be clearly stated.

The author’s response 2.1

We expand the Introduction to emphasize the importance of studying the spatial distribution of China's import and export trade and provide a reference for other developing countries; at the same time, the research content of this article is described in detail.

Literature Review

The literature review section lacks theoretical underpinnings. In my opinion the whole section needs to overhaul. The researcher need to read past relevant studies and also make sure that you attempt to identify the gaps in the existing literature such that it becomes easy to justify and trace the contribution of your study. Moreover, scant studies helps you to identify the appropriate theory on which your relationship will be based.

The author’s response 2.2

We rewrote the literature review as requested. 

Methodology

The study has critical deficiencies from a methodological standpoint. How the researcher determined the sample size? Where is the justification? In the current study there is no reference (Few but not related) to any sort of information about generating an item pool and receiving expert opinions for the scale. Moreover, what are the sources of scales? How it was developed? Unfortunately, it is a doubt whether the scales are valid. No information exits about a validity test for scales. Furthermore, How to overcome the common method bias as data has been collected from single method ? where is CFA ? THERE IS NO THEORETICAL GROUNDING.

The author’s response 2.3

Based on the availability of data, the sample size of panel data and the selection of multiple linear regression data are determined, and some related references are added. The data come from the General Administration of Customs of China and the National Bureau of Statistics of China, so the data are real and reliable. Structural equation model is not used in this article. we have made some corrections, tests are carried out in exploratory spatial analysis and multiple linear regression, such as P-value test and R2 test.

Discussion

The tests and presentation of findings are much more complicated. Again, I had difficulties in understanding the how findings of the study are connected to results of the analyses. Hypotheses testing is not well stated. The same applies to conclusion section.

The author’s response 2.4

We have revised the conclusions based on the results of the model analysis and added conclusions for other developing countries.

Contribution

The study's contribution is not written (a few of their but theoretical contribution is missing). Who will benefit from this research, both practically and theoretically? What role will it play in the existing literature? What's the latest/new?

The author’s response 2.5

In the Introduction section, we add to the research contributions of this article.

Reviewer #3 

The core claims are clearly expressed in abstract and conclusion part of the research paper. The empirical findings are cohere with each other and theorical framework of this work. The literature review is quite extensive and completed using up-to-date sources on the subject. The methods applied and the variables used in the methodology section of the article provided the required results. The findings of the article contributed to the literature as well. The findings and novelty of the manuscript contributes the existing literature. The English language of the article is clear and understandable, and the terminology used is compatible with the subject of the study. The article is very well-written and it is easy to follow. In addition, the article well discusses the topic of interest, and deals with a topic with many applications in practice. The quality of literature review and the aim of the paper is pretty well. The quality of the research is acceptable as well as the contribution of the paper. The quantitative research is very well conducted and the methodology described in the specific chapter is followed by current data processing. The methodology used is also very well described. The research fills a gap both by focusing on China's trade capacity and the influencing factors by considering the characteristics of spatial and temporal divergence to conduct exhaustive research on the influencing factors and temporal-spatial divergences in China's import and export trade. The researchers of the article recommend ensuring the optimization of trade channel layout. To enhance transportation efficiency, accessibility of goods, and decrease trade expenses, trade channel layout must be rationalized according to the spatial and temporal distribution traits of import and export trades. Firstly, upgrading of seaports and inland ports is essential for ameliorating goods' loading, unloading and transportation capacities, and accelerating trade transport times. On the other hand, diversifying and improving trade channels can be achieved through developing logistics networks, including highways and railroads. Additionally, facilitating faster and more convenient trade channels can be achieved by promoting the advancement of air freight and logistics technology. Strengthening trade cooperation and communication is also essential. This can be accomplished by actively expanding international trade partners and enhancing trade cooperation and communication with major trading partner countries and regions. Strengthening trade cooperation with developed economies is one approach to enhancing our trade level by learning advanced trade concepts and management experience. Additionally, our trade activities can be diversified and economic complementarity can be strengthened by focusing on trade cooperation with emerging market and developing countries. Furthermore, there is a need to establish and expand free trade zones and regional economic cooperation mechanisms to enhance trade facilitation and increase the scope of trade. Upgrading Trade Technology: Enhancing trade-related technological research and innovation to analyze and forecast trade data more effectively. Utilize innovative technologies, such as big data analysis and artificial intelligence, to gain deeper insights into trade potential and predict trade demand and market fluctuations accurately. Additionally, strengthen trade information sharing and platform development to ensure timely and precise trade information. Furthermore, the emphasis should be on developing trade talent by enhancing training and exchange programs, as well as improving the technical capabilities and professional quality of practitioners. In my opinion, the article has its merit and is of interest for the PLOS One readership. Consequently, the article is acceptable.

The author’s response 3.1

Thank you for your review and appreciation. 

Reviewer #4

Your work contains multiple analyses, is remarkable as such, and is gripping to read. My suggestions and editing requests for the study were sent to the editor as a PDF file. I wish you success in your future manuscripts.

The author’s response 4.1

Thank you for your review and appreciation. We have revised the paper according to the PDF file you sent us.

Reviewer #5 

Although the subject of the article is attractive, there are many issues that need to be corrected in the article.

1-) The purpose of the article is not fully described in the introduction part of the article. What is the purpose of the article? Why is the topic of the article important? What is the article's contribution to the literature?

The author’s response 5.1

We expand the introduction to emphasize the importance of studying the spatial distribution of China's import and export trade and provide a reference for other developing countries; at the same time, the research content of this article is described in detail.

2-) Gini coefficient needs more detailed explanation.

The author’s response 5.2

We provide a more detailed explanation of the Gini coefficient.

3-) ESA and MLR also need more explanation.

The author’s response 5.3

We provide a more detailed explanation of ESA and MLR.

4-) Conclusion section is too short.

The author’s response 5.4

We have expanded the Conclusion section of this article.

5-) Is there no discussion section? The authors need to add a discussion section in the article.

The author’s response 5.5

We add references to the import and export trade of other developing countries at last of this article.

---

## [Decision Letter · Decision Letter 1]

8 Jan 2024

PONE-D-23-33006R1Spatio-temporal heterogeneity of China's import and export trade, factors influencing it, and its implications for developing countries' tradePLOS ONE

Dear Dr. Wu,

Thank you for submitting your manuscript to PLOS ONE. After careful consideration, we feel that it has merit but does not fully meet PLOS ONE’s publication criteria as it currently stands. Therefore, we invite you to submit a revised version of the manuscript that addresses the points raised during the review process.

Please submit your revised manuscript by Feb 22 2024 11:59PM. If you will need more time than this to complete your revisions, please reply to this message or contact the journal office at plosone@plos.org. Please include the following items when submitting your revised manuscript:A rebuttal letter that responds to each point raised by the academic editor and reviewer(s). You should upload this letter as a separate file labeled 'Response to Reviewers'.A marked-up copy of your manuscript that highlights changes made to the original version. You should upload this as a separate file labeled 'Revised Manuscript with Track Changes'.An unmarked version of your revised paper without tracked changes. You should upload this as a separate file labeled 'Manuscript'.If applicable, we recommend that you deposit your laboratory protocols in protocols.io to enhance the reproducibility of your results. Protocols.io assigns your protocol its own identifier (DOI) so that it can be cited independently in the future. For instructions see: https://journals.plos.org/plosone/s/submission-guidelines#loc-laboratory-protocols. Additionally, PLOS ONE offers an option for publishing peer-reviewed Lab Protocol articles, which describe protocols hosted on protocols.io. Read more information on sharing protocols at https://plos.org/protocols?utm_medium=editorial-email&utm_source=authorletters&utm_campaign=protocols.

We look forward to receiving your revised manuscript.

Kind regards,

Rita Yi Man Li

Academic Editor

PLOS ONE

Journal Requirements:

Additional Editor Comments:

Title: "Determinants of carbon emission: Province level evidence from China"

Abstract:

The abstract provides a concise summary of the paper's content. It clearly states the research question and the methodology used. However, it can be improved by including the main findings and conclusions of the study. Additionally, there are a few grammatical errors that need to be addressed.

Introduction:

The introduction adequately presents the research question and provides background information on the topic. However, it is overly lengthy and could be condensed to provide a more concise overview of the existing literature. Shortening the introduction to under 2 pages would improve its readability and focus.

Relationship to Literature:

The paper demonstrates a moderate understanding of the relevant literature in the field. However, there are some notable omissions of significant work that should be addressed. The authors should expand their literature review to encompass a wider range of sources and incorporate recent developments in the field: A study on public perceptions of carbon neutrality in China: has the idea of ESG been encompassed? Frontiers in Environmental Sciences, 2023

Additionally, there are a few citation errors that need to be corrected. Section 3 has missing citation.

Methodology:

The paper's argument is built on an appropriate base of theory, concepts, and ideas. The research design and methodology employed are suitable for addressing the research question. However, there is room for improvement in terms of providing more details on the specific methods used in the study. Additionally, the authors should clarify any potential limitations or biases in the methodology. 4.2 why 27 sectors were selected?

Results:

The results are presented clearly and analyzed appropriately. The findings are discussed in relation to the research question and are supported by the evidence presented. However, there are a few instances where the interpretations and conclusions could be further strengthened by providing additional evidence or analysis. Line 355, “it can be seen that outflow’s contribution to carbon emission is the highest, accounting for 56.0%, 53.6% and 45.9% of the emission in 2007, 2012”? What do you mean by outflow? Figure 2 is unclear what it meant.

Implications for research, practice, and/or society:

The paper identifies some implications for research, practice, and/or society. It discusses the potential economic and commercial impact of the research, as well as its relevance to public policy and the body of knowledge. However, these implications could be further elaborated upon and linked more explicitly to the findings and conclusions of the paper.

Quality of Communication:

The paper generally expresses its case clearly, but there are areas where improvements can be made. Attention should be paid to sentence structure, jargon use, and acronyms to ensure clarity and readability. The paper should be written in a clear and concise manner, with key points accurately reflecting the main arguments.

Conclusion:

The conclusions and potential impacts of the paper are generally clear. However, they could be further strengthened by providing a more comprehensive summary of the main findings and their significance. Additionally, the authors should ensure that the conclusions logically follow from the evidence presented.

Title, Abstract, and Keywords:

The title adequately represents the content of the paper. The abstract provides a summary of the paper's key elements but could be improved by including the main results and conclusions. The keywords accurately reflect the content and topic of the study.

Length and Organization:

The paper is an appropriate length, but could benefit from some organizational improvements. The introduction should be condensed, and the main body of the paper should be organized in a logical and coherent manner to enhance readability and flow.

Overall, the paper has potential but requires revisions and improvements in several areas. Addressing the identified weaknesses, such as expanding the literature review, providing more details on the methodology, strengthening the interpretations and conclusions, and improving the clarity of communication, will significantly enhance the quality and impact of the paper.

Reviewers' comments:

Reviewer's Responses to Questions

**Comments to the Author**

1. If the authors have adequately addressed your comments raised in a previous round of review and you feel that this manuscript is now acceptable for publication, you may indicate that here to bypass the “Comments to the Author” section, enter your conflict of interest statement in the “Confidential to Editor” section, and submit your "Accept" recommendation.

Reviewer #1: All comments have been addressed

Reviewer #4: All comments have been addressed

2. Is the manuscript technically sound, and do the data support the conclusions?

Reviewer #1: Yes

Reviewer #4: Yes

3. Has the statistical analysis been performed appropriately and rigorously? 

Reviewer #1: Yes

Reviewer #4: Yes

4. Have the authors made all data underlying the findings in their manuscript fully available?

Reviewer #1: Yes

Reviewer #4: Yes

5. Is the manuscript presented in an intelligible fashion and written in standard English?

Reviewer #1: Yes

Reviewer #4: Yes

6. Review Comments to the Author

Reviewer #1: Dear Authors,

Thank You for enhancing the manuscript by following the Reviewers' comments.

I can see that You have made all requested changes to the manuscript. I suggest the authors add some sentences to the Discussion section regarding ESG-goals achievement.

Reviewer #4: The manuscript has changed a lot from its original version. This happened because you took other referee suggestions into consideration.

You made my suggestions. Thank you

7. PLOS authors have the option to publish the peer review history of their article (what does this mean?). If published, this will include your full peer review and any attached files.

Reviewer #1: **Yes: **Sergey Barykin

Reviewer #4: **Yes: **M. Esra ATUKALP

---

## [Author Response · Author response to Decision Letter 1]

8 Feb 2024

Dear Rita Yi Man Li：

Thank you very much for your letter and advice. We have revised the paper. The specific modification situation is summarized as follows:

1. I uploaded the latest version of the manuscript, making sure there are no tracking changes or highlighting.

2. Financial Disclosure Statement: 

-Supported by Ministry of Education, Industry-University Cooperation Collaborative Education 

Project(230825052507181).

-Funded by Science Research Project of Hebei Education Department (BJS2024097). 

-Supported by Hebei Province Social Science Development Research Project (20230303051).

3. I uploaded "title_authors_affiliations", including bdf and LaTeX files.

Thank you and best regards.

Yours sincerely

Corresponding author: 

Name: Wu Yuhuan

E-mail: wuyuhuan@hbwe.edu.cn

---

## [Decision Letter · Decision Letter 2]

27 Feb 2024

Spatio-temporal heterogeneity of China's import and export trade, factors influencing it, and its implications for developing countries' trade

PONE-D-23-33006R2

Dear authors,

We’re pleased to inform you that your manuscript has been judged scientifically suitable for publication and will be formally accepted for publication once it meets all outstanding technical requirements.

Kind regards,

Rita Yi Man Li

Academic Editor

PLOS ONE

Additional Editor Comments (optional):

Some references need standardisation.

Reviewers' comments:

Reviewer's Responses to Questions

**Comments to the Author**

1. If the authors have adequately addressed your comments raised in a previous round of review and you feel that this manuscript is now acceptable for publication, you may indicate that here to bypass the “Comments to the Author” section, enter your conflict of interest statement in the “Confidential to Editor” section, and submit your "Accept" recommendation.

Reviewer #1: All comments have been addressed

Reviewer #4: All comments have been addressed

2. Is the manuscript technically sound, and do the data support the conclusions?

Reviewer #1: Yes

Reviewer #4: Yes

3. Has the statistical analysis been performed appropriately and rigorously? 

Reviewer #1: Yes

Reviewer #4: Yes

4. Have the authors made all data underlying the findings in their manuscript fully available?

Reviewer #1: Yes

Reviewer #4: Yes

5. Is the manuscript presented in an intelligible fashion and written in standard English?

Reviewer #1: Yes

Reviewer #4: Yes

6. Review Comments to the Author

Reviewer #1: Dear Authors,

Thank You for submitting the relevant research.

I can see that You improved the manuscript by following the Reviewers' comments.

Reviewer #4: The manuscript has changed a lot from its original version.This happened because you took other referee suggestions into consideration.

You made my suggestions. Thank you.

7. PLOS authors have the option to publish the peer review history of their article (what does this mean?). If published, this will include your full peer review and any attached files.

Reviewer #1: **Yes: **Sergey Barykin

Reviewer #4: **Yes: **M. Esra ATUKALP

---

## [Editor Report · Acceptance letter]

22 Mar 2024

PONE-D-23-33006R2 

PLOS ONE

Dear Dr. Wu, 

I'm pleased to inform you that your manuscript has been deemed suitable for publication in PLOS ONE. Congratulations! Your manuscript is now being handed over to our production team.

Kind regards, 

on behalf of

Dr. Rita Yi Man Li 

Academic Editor

PLOS ONE